# In Situ Rumen Degradation Characteristics and Bacterial Colonization of Corn Silages Differing in Ferulic and *p*-Coumaric Acid Contents

**DOI:** 10.3390/microorganisms10112269

**Published:** 2022-11-15

**Authors:** Yan-Lu Wang, Wei-Kang Wang, Qi-Chao Wu, Fan Zhang, Wen-Juan Li, Sheng-Li Li, Wei Wang, Zhi-Jun Cao, Hong-Jian Yang

**Affiliations:** State Key Laboratory of Animal Nutrition, College of Animal Science and Technology, China Agricultural University, Beijing 100193, China

**Keywords:** corn silage, rumen-attached bacteria, phenolic acids, in situ degradation

## Abstract

In plant cell wall, ferulic acid (FA) and *p*-coumaric acid (*p*CA) are commonly linked with arabinoxylans and lignin through ester and ether bonds. These linkages were deemed to hinder the access of rumen microbes to cell wall polysaccharides. The attachment of rumen microbes to plant cell wall was believed to have profound effects on the rate and the extent of forage digestion in rumen. The objective of this study was to evaluate the effect of bound phenolic acid content and their composition in corn silages on the nutrient degradability, and the composition of the attached bacteria. Following an in situ rumen degradation method, eight representative corn silages with different FA and *p*CA contents were placed into nylon bags and incubated in the rumens of three matured lactating Holstein cows for 0, 6, 12, 24, 36, 48, and 72 h, respectively. Corn silage digestibility was assessed by in situ degradation methods. As a result, the effective degradability of dry matter, neutral detergent fibre, and acid detergent fibre were negatively related to the ether-linked FA and *p*CA, and their ratio in corn silages, suggesting that not only the content and but also the composition of phenolic acids significantly affected the degradation characteristics of corn silages. After 24 h rumen fermentation, Firmicutes, Actinobacteria, and Bacteroidota were observed as the dominant phyla in the bacterial communities attached to the corn silages. After 72 h rumen fermentation, the rumen degradation of ester-linked FA was much greater than that of ester-linked *p*CA. The correlation analysis noted that *Erysipelotrichaceae_UCG-002*, *Olsenella*, *Ruminococcus_gauvreauii_group*, *Acetitomaculum*, and *Bifidobacterium* were negatively related to the initial ether-linked FA content while *Prevotella* was positively related to the ether-linked FA content and the ratio of *p*CA to FA. In summary, the present results suggested that the content of ether-linked phenolic acids in plant cell walls exhibited a more profound effect on the pattern of microbial colonization than the fibre content.

## 1. Introduction

Corn silage, known as one of the most widely used forage sources for ruminants, has high biomass yield, good palatability, and good fermentation quality [1]. However, the plant cell walls limited the digestion and utilization of forage nutrients by the ruminant host [2]. Plant cell walls are complex systems mainly consisted of polysaccharides, phenolics (monomers and polymers), and proteins [3]. Within plant cell walls, lignin content affecting forage degradation has been studied for decades, and the structure and content of lignin and phenolic acids were believed to have a significant impediment in the digestion of the plant cell wall [4]. Jung and Vogel noted that the degradability of forages varied even though the content of neutral detergent fibre (NDF) was similar [5]. Grabber et al., used a biomimetic model to demonstrate that the structure of ferulate cross-linking in comparison with lignin composition presented a greater negative effect on the rate and extent of digestion of hemicellulose in plant cell walls [6]. Ferulic acid (FA) and *p*-coumaric acid (*p*CA) affecting fibre digestibility were the most abundant phenolic acids in plant cell wall, which can account for up to 80% of the total phenolic acids [7]. As shown in Figure 1, FA can be ester-linked via its carboxylic acid group to the C(O)5 position of the arabinofuranosyl side group attached at the C(O)2 to the xylan chain while *p*CA in plant cell walls are mainly bound to lignin, only a small amount of *p*CA are attached to polysaccharides [8,9]. Lignin and arabinoxylans can be connected by ferulate molecules through ether bonds and form dimeric structures of cross-link arabinoxylan chains to each other and to lignin [10]. The *p*CA in plant cell walls are mainly esterified to the γ-position of phenylpropanoid side chains of S units in lignin [11]. These linkages were believed to work as a barrier to limits the utilization of plant cell walls in the rumen [12,13].

Rumen microbes the carry out microbial fermentation include bacteria, archaea, fungi, and protozoa, and they anaerobically break down forage and the other feeds to produce volatile fatty acids (e.g., acetate, propionate and butyrate), providing 70% metabolic energy of feedstuffs to ruminants [15,16,17]. Within rumen ingesta, 70~80% of microbial organic matter are associated with particle-associated bacteria which play an important role in the utilization of feeds [18]. Bacteria community attached to rumen ingesta varied depending on chemical compositions of feeds [17,19]. A recent metagenomic analysis study revealed that Bacteroidota, Firmiutes, Verrucomicrobiota, and Fibrobacterota were enriched for genes related to the degradation of lignocellulosic polymers and the fermentation of degraded products into short chain volatile fatty acids after six different lignocellulosic biomasses were incubated in rumen-fistulated Taleshi cattle [20]. The inhibition of linked phenolic acids on the digestion of feed in rumen has been reported for years. Previous studies suggested that phenolic acids cross-linkages formed an obstruction through both substitution and steric hindrance for the accessing of hydrolytic enzymes to their polysaccharide substrate [21]. However, it is still unknown how these cross-linkages influence the rumen microbiota attached to forage.

In the present study, representative samples of corn silages with different FA and *p*CA contents were chosen as experimental forage materials. The objective was to evaluate the effect of bound FA and *p*CA in plant cell walls on the rumen degradation as well as the association between chemical composition and solid-attached bacteria diversity using an in situ rumen incubation method. We assumed that the bacteria attached to corn silages with different phenolic acid contents could provide insight for further understanding of phenolic acids degradation in the rumen. 

## 2. Materials and Methods

### 2.1. Preparation of Corn Silage

Eight representative corn silages with different phenolic acids concentration were collected from different farms in He Bei, Shan Dong, Bei Jing, and He Nan provinces of China. All samples were dried in a forced air oven (DHG-9420A, YiHeng Scientific Instrument Limited Company, Shanghai, China) at 65 °C for 48 h and ground (KRT-34, KunJieYu Mechanical Equipment Co., Ltd., Beijing, China) to pass through a 2.00 mm screen, and stored prior to chemical analyses, phenolic acids content and in situ degradation trials.

### 2.2. Sampling and Analytical Methods

Following the methods of the Association of Official Analytical Chemists (AOAC) [22], regarding the corn silage samples, dry matter (DM) was analyzed in oven drying at 105 °C for 4 h. Neutral detergent fibre (NDF) and acid detergent fibre (ADF) were determined following Van Soest’s method [23]. During the assay of NDF, the heat stable amylase used for the remove of starch from the corn silages was purchased from Aladdin (Shanghai, China).

### 2.3. Determination of Extracted Phenolic Acids 

The extraction of total and ester-linked phenolic acids were determined as described by Cao et al. [24]. Briefly, the total and ester-linked phenolic acids were released by 4.0 M and 2.0 M NaOH at 170 °C and 39 °C, respectively.

The standard chemicals of FA and *p*CA were purchased from Sigma-Aldrich Company (St. Louis, MO, USA) and dissolved in methanol into 200, 100, 50, 25, 12.5, 6.25, and 3.125 μg/mL, respectively. The quantification of FA and *p*CA was conducted by HPLC (WuFeng Co., Ltd., Shanghai, China) with a C18 column (250 × 4.6 mm, 5 μm, pH 2–8, Waters, Milford, MA, USA). The binary gradient solvent system contained (A) chromatographic methanol and (B) formic acid in distilled water (0.4%, *v/v*). The gradient started at 36% solvent A, passing to 44% in 5 min, and back to 36% in 20 min. The sample (10 μL) was injected into HPLC loop for the determination. The sample was analyzed at 30 °C with a flow of 1 mL/min at a wavelength of 320 nm. The FAeth content in forages was calculated as the difference between total FA and ester-linked FA [8].

### 2.4. In Situ Rumen Incubation 

Three lactating Holstein cows in late-lactation phase with rumen-cannulated were served as experimental animals. The cows were arranged in three stalls and fed twice daily at 8:00 am and 15:00 pm with 2.75 kg of imported Alfalfa, 12.25 kg of corn silage, and 10 kg of a commercial concentrate per animal, and free access to water. All animals in this experiment were conducted in accordance with the Institutional Animal Care Committee and the Animal Welfare Guidelines of China Agricultural University (CAU20171014-1).

Approximately 5 g grounded sample of each corn silage was weighted into nylon bags (10 × 20 cm; 50 μm pore size), and these bags were placed into rumens and incubated for periods of 0, 6, 12, 24, 36, 48, and 72 h. All nylon bags (2 bags per corn silage sample for per cow) were simultaneously placed into rumen of each of the three Holstein cows at 08:00 am. When the incubation of each time was completed, all nylon bags were removed from rumen. Regarding the bags collected at 24 h, one of two bags used to recover corn silage residue for subsequent bacterial community analysis. All other bags were rinsed and manipulated in tap water until the water ran clear, then squeezed to remove water and dried at 65 °C for 48 h to determine in situ DM, NDF, ADF, acid detergent lignin (ADL), and phenolic acids degradation. 

### 2.5. Bacterial Community Analysis

#### 2.5.1. Collection of Solid Attached Microbiota and DNA Extraction

Regarding each sample of 8 corn silages, the bags collected for solid attached microbiota assays after 24 h incubation in the rumen were washed thoroughly with distilled water for 3 times to remove the loosely attached microbes and then squeezed the bags using sterile gloves to remove excess water [19]. Finally, the residue solid contents were stored in sterile centrifuge tubes, then removed into liquid nitrogen, and frozen at −80 °C. Following the attached bacteria collection method as described Larue et al. [25], the solid attached microbiota on corn silage residues were subsequently stripped in a dissociation buffer (storage for 24 h at 4 °C before using) with 0.1% (*v/v*) Tween 80, 1% (*v/v*) methanol, and 1% (*v/v*) tertiary butanol (adjusted to pH 2.0) on ice [26]. All the samples were vortexed with a vortex meter (Vortex-2, Shanghai Huxi Industry Co., Ltd., Shanghai, China). Then the samples were centrifuged at 500× *g* to remove the silage particle. The supernatants containing the microbial cells weremoved into new sterile containers. And these steps were repeated three times. The collected filtrate was centrifuged at 12,000× *g* and 4 °C for 10 min to obtain the sediment of microbial cells. The extraction of total DNA used an OMEGA Stool DNA Isolation Kit (MoBio Laboratories, Carlsbad, CA, USA). The quality and concentration of DNA were calculated using NanoDrop 1000 spectrophotometry.

#### 2.5.2. PCR Amplification, Illumina Sequencing of 16S rRNA Gene, and Sequence Analysis

The V3-V4 hypervariable regions of bacterial 16S rRNA gene were amplified with the primers 515F (5′-GTGCCAGCMGCCGCGGTAA-3′) and 806R (5′-GGACTACHVGGGTWTCTAAT-3′) [27]. A two stage PCR amplification was performed in triplicate using an ABI GeneAmp^®^ 9700 (Applied Biosystems, Waltham, MA, USA). Each 25 μL PCR contained 12.5 μL 2 × PCR Master Mix, 5.5 μL nuclease-free water, 3 μL BSA, 2 μL template DNA, 2 μL forward primer (5 μM). The PCR condition consisted of an initial denaturation at 95 °C for 5 min followed by 32 cycles of 95 °C for 45 s, 55 °C for 50 s, and 72 °C for 45 s, with a final extension at 72 °C for 10 min. The PCR amplicons were purified using a QIAquick Gel Extraction Kit, quantified using Real-Time PCR (ABI GeneAmp^®^ 9700), and sequenced at Majorbio Company in Shanghai. 

After the PCR, image analysis, base calling, and error estimation were performed using Illumina Analysis Pipeline Version 2.6 for further data analysis. The sequencing data were submitted into the Uparse software (version 7.0.1090). By classification operations, sequences are classified according to their similarity. Usually, the sequence was clustered into operational taxonomic units (OTU) division at 97% similarity cut-off using Uparse (V11.0.667) [28]. The taxonomies were assigned to each OTU using the rdp Classifier tool (V2.1) by applying a minimum confidence value of 0.7 [29]. The community composition was counted on domain, phylum, class, order, family, genus, and species levels. The rarefaction curves and alpha diversity indices were calculated using the core_diversity_analyses.py script in the QIIME pipeline. Rarefaction plots and alpha diversity indices, including Good’s coverage, Shannon, Simpson, and Chao1 index were created using Mothur software (V1.30.2).

### 2.6. Calculations and Statistical Analysis

Regarding different incubation time (t), in situ digestion rates *(DRt*) of DM, NDF, ADF, *p*CAest and FAest in corn silages was calculated using Equation (1):(1)$$DRt=\left(W_{i}-W_{r}\right)/W_{i}\;$$
where W_i_ is the initial mass weight before incubation and W_r_ is the recovered mass weight from nylon bags after the incubation of time t.

To obtain kinetic degradation constants of each corn silage in terms of DM, NDF, ADF, *p*CAest, and FAest, DRt values were fitted to an exponential model [30] using the nonlinear (NLIN) procedure of the statistical software package SAS (version 9.4, SAS Institute Inc., Cary, NC, USA) as below Equation (2):(2)$$DRt=a+b\;\;\left(1-e^{-c\times time}\right)$$
where *a* represents the immediately soluble fraction, *b* represents the insoluble but potentially degradable fraction, *c* represents the rate constant for the degradation per unit time, and *e* is the base of a natural logarithm. 

The effective degradabilities (ED) of DM, NDF, ADF, *p*CAest and FAest were estimated using Equation (3) [30]:(3)$$ED=a+\left[\frac{bc}{c+k}\right]$$
where *k* presents the ruminal solids outflow rate, which was assumed as 0.0253/h [31]. 

After the above kinetic constant estimation, the pressent study resulted 8 corn silages × 3 cows = 24 degradation constants for each of DM, NDF, ADF, *pCA*est and FAest. The original contents of NDF, ADF, *p*CAest, *p*CAeth, FAest, FAeth, *p*CA/FA, ADL in 8 corn silages were then subjected to correlationship analysis with the degradation constants and bacterial community attached to corn silages after 24 h incubation time using the correlation (CORR) procedure of the software package SAS for windows (version 9.4, SAS Institute Inc., Cary, NC, USA). Significance was declared at *p* < 0.05, unless otherwise noted.

## 3. Results

### 3.1. Chemical Composition and Rumen Degradation of Corn Silages

As shown in Table 1, the NDF content of corn silages, ranging from 377.40 to 531.43 g/kg DM, was the predominant nutrient in corn silages. The highest and lowest NDF contents occurred in WCS8 and WCS1, respectively. The concentrations of the esterified and etherified fractions of *p*CA (6.08 to 8.28 g/kg DM; 0.69 to 2.06 g/kg DM) and FA (2.90 to 5.45 g/kg DM; 0.08 to 2.16 g/kg DM) varied widely among eight corn silages.

The dynamic rumen degradation of DM, NDF, and ADF of corn silages were shown in Figure A1. WCS 1 presented greater DM, NDF, and ADF disappearance than other groups after 72 h of rumen incubation (*p* < 0.05). In situ degradation characteristics of DM, NDF, and ADF were listed in Table A1. The DM and NDF rate constant for the degradation of fraction b (c) ranged from 0.047 to 0.088 h^−1^ and from 0.019 to 0.027 h^−1^, respectively. 

The dynamic rumen degradation of *p*CAest and FAest of corn silages were presented in Figure A2. After 72 h of rumen incubation, the disappearance of FAest was remarkably greater than that of *p*CAest in all corn silage samples. The degradation characteristics of *p*CAest and FAest were listed in Table A2. The ED value of FA ranged from 44.25 to 66.92, which was much higher than that of *p*CA. The immediately soluble fraction (a) of *p*CAest (11.18% to 13.18%) was lower than that of FAest (18.12% to 22.15%).

### 3.2. Relationships of Cell Wall Composition with Rumen Degradation

The correlations between phenolic acid contents and in situ degradation characteristics were presented in Table 2. The concentration of FAeth presented highest correlation with the ED value of DM (r = −0.91, *p* < 0.01) while the *p*CAeth content presented highest correlation with the ED value of NDF (r = −0.92, *p* < 0.01) compared with the other fractions in plant cell walls. The immediately soluble fraction (a) and the insoluble but potentially degradable fraction (b) constants of DM and NDF were negatively correlated with *p*CAeth and FAeth contents. Meanwhile, the FAest content in corn silage was positively correlated with the immediately soluble fraction and the insoluble but potentially degradable fraction constants of DM, NDF, and ADF.

### 3.3. 16S rRNA Gene Sequencing

The paired-end sequencing of PCR amplicons from V3-V4 region of the 16S rRNA gene resulted in 1,894,344 raw reads and an average 59,198 sequences per sample with a length greater than 300 bp. After quality trimming, the sequences were merged into 1,844,539 sequences with an average length 414 bp and an average of 57641 sequences per sample. 

### 3.4. Diversity of the Bacterial Microbiota Attached to Different Corn Silages after 24 h of Rumen Fermentation

According to the Figure A3, the Good’s coverage in all samples was greater than 0.99 which indicated that the 24 h rumen incubation provided sufficient OTU coverage for later analysis of bacterial composition. From Table 3, the Pearson correlation coefficients between the α diversity of attached bacteria and NDF, ADF, and phenolic acids were listed. There was no significant correlation between α diversity and the initial content of *p*CA. The initial content of FAest presented a negative effect on Chao 1 values, Shannon and Ace index, while the FAeth content was positively correlated with these values (*p* < 0.05). 

### 3.5. Community Composition of Corn Silage-Attached Microbes 

In the present study, a total of 18 bacterial phyla were identified from forage-attached microbial communities. Firmicutes (65.05% to 79.85%), Bacteroidetes (4.40% to 23.52%), and Actinobacteriota (6.11% to 15.81%) were the three dominant phyla. The lowest Firmicutes was observed in WCS8 which had the highest lignocellulose and FAeth contents. At the family level (Figure 2b), a total of 112 families were detected from the forage-attached microbial communities. Lachnospiraceae (27.74% to 36.85%), Erysipelatoclostridiaceae (1.53% to 20.48%), Prevotellaceae (3.35% to 17.95%), and Acidaminococcaceae (4.35% to 12.74%) were the dominant families. The dominant genus were *Erysipelotrichaceae_UCG-002* (1.06% to 19.72%), *Lachnospiraceae_Nk3A20_group* (8.88% to 14.38%), *Succiniclasticum* (4.35% to 11.98%), and *Prevotella* (2.57% to 13.24%). 

### 3.6. The Correlationships between Attached Microbiota and Corn Silage Cell Wall Contents

The Pearson correlation results between corn silage cell wall contents and the community composition of rumen-attached microbiota were listed in Table 4. At the phyla level, Firmicutes was negatively related to the content of NDF (r = −0.37, *p* < 0.05) and ADL (r = −0.36, *p* < 0.05). The abundance of Actinobacteriota was negatively related to the FAeth (r = −0.73, *p* < 0.01), while Bacteroidota was positively correlated with NDF, ADF, ADL, FAeth the *p*CA/FA ratio. At the family level, Prevotellaceae was positively related with corn silage contents of *p*CAeth, FAeth as well as the *p*CA/FA ratio. Erysipelatoclostridiaceae, Bifidobacteriaceae, and Atopobiaceae were negatively corelated with FAeth. At the genus level, the FAeth in corn silage was negatively related to *Erysipelotrichaceae_UCG-002* (r = −0.74, *p* < 0.01), *Olsenella* (r = −0.68, *p* < 0.01), *Ruminococcus_gauvreauii_group* (r = −0.51, *p* < 0.05), *Acetitomaculum* (r = −0.59, *p* < 0.01), and *Bifidobacterium* (r = −0.50, *p* < 0.05). Meanwhile, a negative correlation occurred between the FAest content in corn silage and *Prevotella* at the genus level. The abundance of *Prevotella* was positively correlated to *p*CAest, *p*CAeth, FAeth, and *p*CA/FA ratio. The *p*CAeth and *p*CA/FA ratio of corn silages were positively correlated with *Prevotella*. The initial contents of NDF, ADF, and ADL were both negatively correlated with *Erysipelotrichaceae_UCG-002* and *Acetitomaculum* but positively correlated with *Prevotella.*

## 4. Discussion

### 4.1. Chemical Composition of Corn Silages 

The FA and *p*CA are the most abundant hydroxycinnamic acids in plant cell walls, and they are highly involved in the lignification process during plant cell wall development [11,32]. In corn silages, the ester-linked FA and *p*CA were much greater than their ether forms. The *p*CA/FA ratio was greater than 1, suggesting that corn silages contained more *p*CA than FA contents. In a previous study, similar phenolic acid profile as described in the present study were also reported in both conventional and brown midrib corn silages [33]. 

### 4.2. The Relationships between the Contents of Phenolic Acids and Digestibility

Hartley et al., noted that *p*CAest had a negative effect on forage digestibility, and the digestibility of the cell wall from ryegrass had a highly significant correlation with the FA/*p*CA ratio [34]. As reviewed in a previous study, increasing studies revealed that both lignin and phenolic acids were the main factors limiting the use of energy by rumen microbes [35]. However, the effect of phenolic acid contents in different plant materials on the utilization of forage is still controversial. In the present study, significant positive correlations were found between FAest and IVDMD, IVNDFD and IVADFD after 72 h rumen incubation. Such a positive correlation between FAest and IVNDFD was also reported in both meadow and Bermuda grass hays [36,37]. The deposition of ester-linked FA always accompanied incorporation of other cell wall components and occurred during primary plant cell wall [38], and this could explain why the positive correlations in the present study occurred between FAest and forage utilization [36]. 

The *p*CA in plant cell wall mainly linked to lignin, and only a small amount of *p*CA were attached to polysaccharides. Therefore, Giada pointed that *p*CA could be considered as an indicator of lignification [39]. In the present study, the content of *p*CAest was negatively related to the ED of ADF in accordance with the result of Argillier [40]. During the second cell wall development, the *p*CA was incorporated into the lignin polymer through ether bounds [41], and this could explain the high correlation coefficient between *p*CAeth and the ED of rumen degradation in the present study. 

With the development of corn maturity, the ester-linked FA was further linked with lignin by ether bonds [24,42]. In the present study, the FAeth content was found negatively correlated to IVDMD and IVADFD as described in previous studies [14,43]. Additionally, a strong negative correlation between the FAeth content and IVNDFD was found in smooth bromegrass, and thus the influence of FAeth was believed independent of the lignin in the fibre fraction [4]. An occasional negative correlation between FAeth and the 48 h IVNDFD also has been observed by Jung et al. [5]. The FAeth content was considered as be an indicator of the cross-linking structure in forages [44,45], and these linkages could protect the cell wall carbohydrates from the adherence of microbes [14,46]. The above could explain why the FAeth content always presented a negative effect on the ruminal degradation of corn silages in the present study. 

### 4.3. In Situ Ruminal Release of Phenolic Acids

Rumen microbes, especially bacteria and fungi, can break down the ester linkages within plant cell walls by secreting feruloyl and *p*-coumaroyl esterase, resulting in the release of free FAs and *p*CA and the improvement of cell wall digestibility [8]. However, due to the presentation of lignin polymer, it is believed somewhat difficult for rumen microbes to release the ether-linkaged FA in an anaerobic environment [47]. Thus, the authors in the present study only detected the degradation of ester-linked phenolic acids during rumen fermentation. The degradation of FAest and *p*CAest after 72 h of rumen incubation ranged from 58.40% to 84.55% and 39.01% to 50.09%, respectively, and this was consistent with Cao’s study that the FAest disappearance in crop residues ranged from 28% to 90%, while the disappearance rate of *p*CAest ranged from 14% to 86% [24]. After in situ rumen incubation, the release of FAest and *p*CAest in mature alfalfa in sheep was 0.84 and 0.62, respectively [48]. The ester-linked FA mainly deposited in the primary cell wall, while the ester-linked *p*CA was in the secondary plant cell wall. A previous study noted that rumen fungi exhibited greater activity of ferulyol esterase in comparison with *p*-coumaroyl esterase [49]. Thus, FAest was more extensively degraded compared with *p*CAest in the present study. In accordance with the results obtained in the present study, Cao et al., also previously noted the FAeth content was negatively correlated with the release of FAest and *p*CAest and the ratio of *p*CA to FA was negatively correlated with the digestibility of ester-linked FA and *p*CA [24]. The above could be explained by the fact that the cross-linking structures prevented microorganisms and microbial enzymes from attacking the cell wall tissues.

### 4.4. Rumen Microbial Colonization of Corn Silages with Different Phenolic Acid Contents

Rapid and simple assay for feruloyl and *p*-coumaroyl esterases has been well built [50]. Previous studies proved that the hydrophobicity of phenolic acids could impede the access of microbes. In addition, ester-linked feruloyl and *p*-coumaroyl groups could inhibit the growth of ruminal bacteria such as *R. flavefaciens* FD1, *Seknomonas ruminantium* HD4, and *Butyrivibrio fibrisolvens* 49 [13,51]. However, limited information is available on the structure, diversity, and density of the attached microbiota of corn silages with different phenolic acid contents. 

Rarefaction analysis reflects richness and relative abundances of species. Shannon and Simpson are traditional indices to describe the diversity of communities [52]. Vahidi et al., reported that the alpha diversity showed no biologically significant differences among the different forages [50]. Gharechahi et al., suggested that the high lignocellulose content of forages limited the species diversity of rumen microbiota [19]. Even though there was no biologically significant correlation between the lignin content and the diversity of communities of microbes after 24 h rumen incubation, the data in Table 3 showed that the corn silage content of ADF presented a positive tendency to Chao and Shannon indices. One possible explanation for these results, speculated by the authors, is the lignin fractions could be degraded during rumen incubation. Thus, compared with the lignin content, the FAeth in plant cell walls seems to have a profound effect on the species richness of the adherent bacteria.

In the present study, the FAeth content was positively correlated with the Chao1, Simpson and ace indices. This result suggested that the ferulate molecules cross-linked with arabinoxylans and lignin and have more profound effects than lignin content itself on the adherence of bacteria on forages.

In the present study, corn silage-attached microbial communities were affiliated to 18 phyla and *Firmicutes*, *Bacteroidetes*, and *Actinobacteriota* took over 98.5% of the phyla in all groups. This result was in accordance with Javad’s and Vahidi’s studies in which Firmicutes and Bacteroidetes were observed as the dominant phyla attached to wheat straw, rice straw, and alfalfa hay in both cow and sheep [19,50,53]. Bacteroidota and Firmicutes were enriched for genes related to the degradation of lignocellulosic polymers [20] while Bacteroidota was found to have highly saccharolytic activities [19]. Within the phylum, members of the families Prevotellaceae displayed significant differences after 24 h rumen incubation. 

### 4.5. Relationships between Plant Cell Wall Contents and Forage-Attached Microbial Communities

In the present study, the authors investigated the microbial colonization on corn silages with different initial phenolic acid contents after 24 h rumen incubation. It has been reported that the forage type could affect the forage-attached microbial community [50]. At the phyla level, Firmicutes was found negatively correlated to NDF and ADL of corn silage in the present study. In Vahidi ’s study, Firmicutes tended to be less associated with fibre-rich feed [50]. The negative correlation of Actinobacteriota negatively with the FAeth content in corn silages implicated that the linkages between lignin and polysaccharides limited the adherence of Actinobacteriota. At the family level, a negative correlation between Bifidobacteriaceae and FAeth was also found in this study. Thoetkiattikul et al., reported that the fibrolytic and cellulolytic bacteria such as Lachnospiraceae and Ruminococcaceae were more abundant in cows fed a high-fibre diet [54]. However, in the present study, Ruminococcaceae was only positively correlated with the FAeth content, while the fibre and phenolic acid did not affect *Lachnospiraceae* enrichness, implicating that the linkages in plant cell wall did not limit the adherence of Ruminococcaceae on the feed fragments. The abundance of Christensenellaceae was negatively related to the *p*CAeth and *p*CA/FA ratio. As previously stated, *p*CA could be incorporated into lignin through ether bonds during the development of the second plant cell walls [41]. Thus, the author speculated that the lignification degree could limit the adherence of Christensenellaceae.

At the genus level, *Bifidobacterium*, *Pseudoscardovia,* and *Syntrophococcus* tended to attach the corn silages with less FAeth contents. *Erysipelotrichaceae_UCG-002*, belonging to Firmicutes, was negatively related to the content of FAeth, ADF, and NDF. Therefore, the *Erysipelotrichaceae_UCG-002* tended to attach to silages with less FAeth and NDF content. *Lachnospiraceae_NK3A20_group* and *Succiniclasticum* were also the indispensable genera in the rumen [55,56]. *Succiniclasticum* acted as a starch-fermenting bacterium and converted succinate to propionate [55,57]. In previous studies, *Succiniclasticum* were more abundant in high-grain diets [58,59]. However, according to our results, the proportions of *Succiniclasticum*, *Lachnospiraceae_NK3A20_group,* and *Christensenellaceae_R-7_group* were not impacted by the phenolic acid and lignin contents. *Christensenellaceae* are beneficial to heath and mainly affected by the host’s genetics [60,61]. Thus, the abundance of *Christensenellaceae_R-7_group* in the present study showed no significantly biological significant correlations with the contents of phenolic acids. *Prevotella* were positively related to the contents of *p*CAeth, FAeth, and ADL and the *p*CA/FA ratio. *Prevotella* presented high hemicellulolytic and proteolytic activities and are important for the degradation of noncellulosic plant polysaccharides and protein in the rumen [54,62]. Recent studies also revealed that *Prevotella* were involved in the digestion of xylan and pectin [54,63] and its abundance was greater in hay-fed goats than in a high-grain group [56]. The results that the content of lignin in forages was positively correlated with the abundance of *Prevotella* in the present study implicated that *Prevotella* might be helpful in the degradation of corn silages.

The occurrence of *Olsenella, Ruminococcus_gauvreauii_group, Acetitomaculum,* and *Bifidobacterium* also were previously reported in the rumen [60,64,65,66]. In the present study, the abundance of these bacteria was found negatively correlated with the content of FAeth in corn silages, implicating that the initial linkages between lignin and arabinoxylans might inhibite the adherence of these bacteria on feed particles. *Olsenella* have been found in the human oral cavity, pig feces, and bovine rumen [67,68] and was believed to play a major role in the degradation of plants [69]. Kraatz reported that *Olsenella* in the rumen had *β*-glucosidase activity to break down glucose [70]. In the present study, the authors found that the abundance of *Olsenella* was mainly affected by the linkages in plant cell walls. *Ruminococcus_gauvreauii_group* are gram-positive obligate anaerobes and can produce acetate as the main end-product of glucose fermentation [58,71]. The negative correlations between *Ruminococcus_gauvreauii_group* and the FAeth content implicated that the linkages in corn silage cell walls could be the main hindrance for the adherence of *Ruminococcus_gauvreauii_group* in feed particles. *Acetitomaculum* can utilize monosaccharides to produce acetate, which was enriched in the long-term feeding of a high-concentrate diet to goats [72]. In addition, Wang et al., also found that the abundance of *Acetitomaculum* increased with the added dietary energy [66]. *Bifidobacterium* was probiotics that can improve gut health, and they were the putative primary degraders of resistant starch [73,74]. The results in the presnt study implicated that the adherence of *Bifidobacterium* on particles was mainly affected by the FAeth content in corn silages.

The present study suggested that the bound phenolic acid contents in addition to the cellulose content in corn silage also affected the forage-attached rumen microbiota. The cross-linkages formed by phenolic acids were an obstruction, through substitution and steric hindrance, in accordance with the previous reports that they would prevent hydrolytic enzymes from accessing their polysaccharide substrates [12,21]. In addition, the linkages in plant cell walls had a more profound effect on the bacteria adhesion, and this could indirectly explain why the negative correlations occurred between the phenolic acid contents and the degradation of corn silages.

## 5. Conclusions

In corn silages, the content of *p*CA was much greater than that FA, especially in the ester-linked form in comparison with the ether-linked form. However, the contents of FAeth and *p*CAeth as well as the *p*CAest content presented negative effects on the fibre degradation of corn silages in the rumen. Meanwhile, the results obtained in the present study demonstrated that the initial contents of phenolic acids in corn silage cell walls could affect the microbial attachments and then further influenced the lignocellulose degradation in the rumen. Taken together, the phenolic acid contents among cross-linkages in plant cell walls may have a more profound effect on the composition of the attached bacteria than the cellulose contents themselves, as found in corn silages.

## Figures and Tables

**Figure 1 microorganisms-10-02269-f001:**
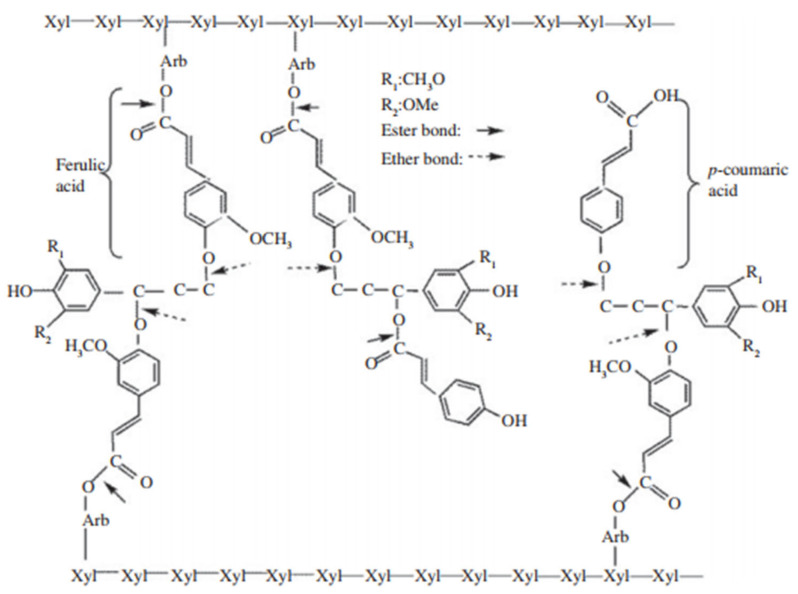
The ester- and ether-linked ferulic acid and *p*-coumaric acid in plant cell walls [14].

**Figure 2 microorganisms-10-02269-f002:**
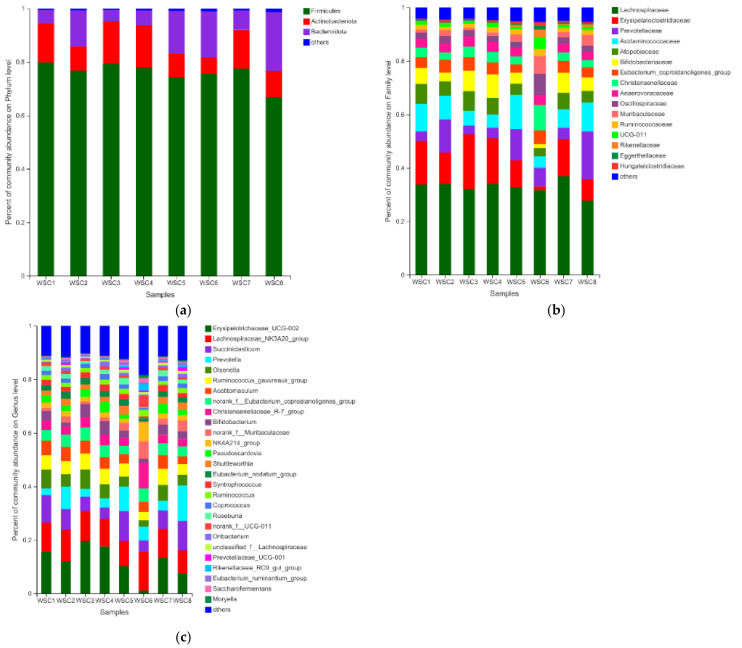
The relative abundance of taxa in rumen microbiota attached to forages. (**a**) The relative abundance of taxa (phyla level) represented in microbial communities attached to eight kinds of corn silages after 24 h of rumen incubation; (**b**) The relative abundance of taxa (family level) represented in microbial communities attached to eight kinds of corn silages after 24 h of rumen incubation; (**c**) The relative abundance of taxa (genus level) represented in microbial communities attached to eight corn silages after 24 h of rumen incubation.

**Table 1 microorganisms-10-02269-t001:** Chemical compositions of corn silages used for in situ rumen incubation.

Item ^1^	WCS1	WCS2	WCS3	WCS4	WCS5	WCS6	WCS7	WCS8
DM (g/kg FM)	97.0	95.6	95.5	91.9	94.8	94.3	94.5	98.5
NDF (g/kg DM)	377.4	378.6	407.5	452.8	477.3	492.9	511.3	531.4
ADF (g/kg DM)	197.9	270.3	225.7	283.0	271.4	283.5	312.1	301.1
ADL (g/kg DM)	14.2	16.6	22.5	29.4	29.2	31.3	37.5	45.5
Phenolic acids (g/kg DM)								
*p*CAest	6.08	8.21	7.91	7.39	6.41	7.1	7.36	8.23
*p*CAeth	0.69	1.80	0.92	1.42	1.44	1.22	2.17	2.06
FAest	5.45	3.63	5.1	4.01	3.33	3.61	3.57	2.9
FAeth	0.08	1.47	0.51	1.17	1.47	1.94	1.29	2.16
*p*CA/FA ratio	1.22	1.96	1.57	1.70	1.64	1.50	1.83	2.03

^1^ FW, fresh weight; DM, dry matter; NDF, neutral detergent fibre; ADF, acid detergent fibre; ADL, acid detergent lignin; FAest, ester-linked ferulic acid; *p*CAest, ester-linked *p*-coumaric acid; *p*CAeth, ether-linked *p*-coumaric acid; FAeth, ether-linked ferulic acid; *p*CA/FA ratio, the ratio of *p*-coumaric acid to ferulic acid.

**Table 2 microorganisms-10-02269-t002:** Pearson correlation coefficients among the major investigated variables and among all corn silages.

Constant ^1^	Phenolic Acid and Lignin Concentrations ^2^							
	*p*CAest	*p*CAeth	FAest	FAeth	*p*CA/FA	NDF	ADF	ADL
DM								
a	−0.75 *	−0.78 *	0.73 *	−0.75 *	−0.83 **	−0.62	−0.70	−0.73 *
b	−0.66	−0.86 **	0.85 **	−0.79 *	−0.89 **	−0.68	−0.83 *	−0.77 *
c (h^−1^)	−0.02	−0.46	0.62	−0.55	−0.38	−0.64	−0.58	−060
a + b	−0.74 *	−0.86 **	0.83 *	−0.80 *	−0.90 **	−0.68	−0.80 *	−0.78 *
ED	−0.57	−0.86 **	0.89 **	−0.91 **	−0.78 *	−0.76 *	−0.84 *	−0.78 *
NDF								
a	−0.68	−0.87 **	0.80 *	−0.66	−0.96 **	−0.54	−0.84 **	−0.64
b	−0.20	−0.55	0.80 *	−0.74 *	−0.53	−0.88 **	−0.81 *	−0.84 **
c	−0.52	−0.66	0.29	−0.36	−0.51	0.03	−0.15	−0.13
a + b	−0.2	−0.43	0.61	−0.57	−0.45	−0.61	−0.53	−0.58
ED	−0.61	−0.92 **	0.77 *	−0.71 *	−0.87 **	−0.57	−0.73 *	−0.70 *
ADF								
a	−0.34	−0.78 **	0.80 **	−0.85 **	−0.58	−0.71 *	−0.60	−0.75 *
b	−0.20	−0.69	0.922 **	−0.91 **	−0.55	−0.72 *	−0.86 **	−0.64
c	−0.36	0.02	−0.45	0.53	−0.17	0.50	0.39	0.90 **
a + b	−0.16	−0.67	0.91 **	−0.92 **	−0.50	−0.72 *	−0.83 *	−0.63
ED	−0.78 *	−0.87 **	0.73 *	−0.73 *	−0.89 **	−0.53	−0.73 *	−0.67
*p*CAest								
a	−0.40	−0.64	0.67	−0.83 *	−0.45	−0.38	−0.41	−0.35
b	−0.49	−0.60	0.71 *	−0.72 *	−0.60	−0.54	−063	−0.54
c	0.04	−0.19	0.09	0.04	−0.18	−0.10	−0.23	−0.16
a + b	−0.55	−0.64	0.63	−0.73 *	−0.56	−0.31	−0.45	−0.34
ED	−0.58	−0.83 *	0.69	−0.63	−0.81	−0.56	−0.72 *	−0.69
FAest								
a	−0.44	−0.75 *	0.92 **	−0.83 *	−0.77 *	−0.73 *	−0.91 **	−0.71 *
b	−0.69	−0.60	0.56	−0.63	−0.64	0.16	−0.45	−0.19
c	−0.05	−0.05	0.14	0.15	−0.33	−0.30	−0.40	−0.33
a + b	−0.67	−0.77 *	0.78 *	−0.76 *	−0.82 *	−0.40	−0.72 *	−0.43
ED	−0.50	−0.57	0.73 *	−0.58	−0.75 *	−0.59	−0.86 **	−0.59

* *p* < 0.05; ** *p* < 0.01. ^1^ DM, dry matter; NDF, neutral detergent fibre; ADF, acid detergent fibre; a, immediately soluble fraction; b, insoluble but potentially degradable fraction; a + b, potential degradation; c, rate constant for the degradation of fraction b; ED, effective degradability. ^2^ ADL, acid detergent lignin; *p*CAest, ester-linked *p*-coumaric acid; *p*CAeth, ether-linked *p*-coumaric acid; FAest, ester-linked ferulic acid; FAeth, ether-linked ferulic acid; *p*CA/FA ratio, the ratio of *p*-coumaric acid to ferulic acid.

**Table 3 microorganisms-10-02269-t003:** Pearson correlation coefficients between the α diversity of the attached bacteria after 24 h and the phenolic acid composition.

Item	Phenolic Acid and Lignin Concentrations ^1^							
	NDF	ADF	ADL	*p*CAest	*p*CAeth	FAest	FAeth	*p*CA/FA
Chao1 index	0.55	0.76	0.49	0.12	0.66	−0.88 **	0.85 **	0.55
Simpson	−0.57	−0.72	−0.48	−0.05	−0.53	0.80 *	−0.85 **	−0.38
Shannon	0.59	0.72	0.48	0.04	0.39	−0.76 *	0.86 **	0.29
Ace	0.54	0.74	0.51	0.19	0.7	−0.87 *	0.84 **	0.6

* *p* < 0.05; ** *p* < 0.01. ^1^ DM, dry matter; NDF, neutral detergent fibre; ADF, acid detergent fibre; FAest, ester-linked ferulic acid; *p*CAest, ester-linked *p*-coumaric acid; *p*CAeth, ether-linked *p*-coumaric acid; FAeth, ether-linked ferulic acid; *p*CA/FA ratio, the ratio of *p*-coumaric acid to ferulic acid.

**Table 4 microorganisms-10-02269-t004:** Pearson correlation coefficients between the α diversity of the attached bacteria after 24 h and the phenolic acid composition.

Item	Phenolic Acid and Lignin Concentrations ^1^							
	NDF	ADF	*p*CAest	*p*CAeth	FAest	FAeth	*p*CA/FA	ADL
Phylum level								
Firmicutes	−0.37 *	−0.34	−0.15	−0.25	0.29	−0.24	−0.17	−0.36 *
Actinobacteriota	−0.28	−0.26	−0.01	−0.23	0.52 *	−0.73 **	−0.14	−0.22
Bacteroidota	0.55 *	0.51 *	0.3	0.51	−0.75 **	0.88 **	0.48 *	0.50 *
Family level								
Lachnospiraceae	−0.18	0.03	−0.12	0.06	0.01	−0.11	0.01	−0.22
Erysipelatoclostridiaceae	−0.55 *	−0.47 *	0.04	−0.27	0.62 **	−0.75 **	−0.13	−0.43 *
Acidaminococcaceae	−0.04	−0.11	−0.27	0.1	−0.19	0.04	0.09	−0.05
Prevotellaceae	0.3	0.35	0.3	0.46 *	−0.66 **	0.63 **	0.57 **	0.34
Bifidobacteriaceae	−0.13	−0.1	0.1	0	0.36	−0.50 *	0.04	0.01
Atopobiaceae	−0.52 *	−0.46 *	−0.02	−0.27	0.58 **	−0.66 **	−0.2	−0.43 *
Eubacterium_coprostanoligenes_group	−0.26	−0.04	0.35	−0.14	0.17	−0.01	−0.02	−0.24
Christensenellaceae	0.18	0.08	−0.1	−0.27	−0.05	0.3	−0.32	0.05
Anaerovoracaceae	−0.31	−0.15	0.06	−0.28	0.15	−0.08	−0.12	−0.33
Oscillospiraceae	0.25	0.19	−0.02	−0.12	−0.2	0.45 *	−0.19	0.13
Muribaculaceae	0.47 *	0.39 *	0.01	0.16	−0.50 *	0.65 **	0.08	0.37 *
Ruminococcaceae	0.3	0.15	−0.2	−0.16	−0.26	0.41 *	−0.18	0.2
Genus level								
* Erysipelotrichaceae_UCG−002*	−0.57 *	−0.48 *	0.03	−0.27	0.62 **	−0.74 **	−0.13	−0.45 *
* Lachnospiraceae_NK3A20_group*	−0.14	−0.02	0.01	−0.15	0.03	0.11	−0.15	−0.21
* Succiniclasticum*	0.06	−0.07	−0.19	0.13	−0.19	0.07	0.11	0.07
* Prevotella*	0.35	0.40 *	0.38 *	0.50 *	−0.65 **	0.63 **	0.60 **	0.42 *
* Olsenella*	−0.46 *	−0.49 *	−0.07	−0.28	0.60 **	−0.68 **	−0.23	−0.37 *
* Ruminococcus_gauvreauii_group*	−0.35	−0.22	0.03	−0.03	0.35	−0.51 *	0.02	−0.27
* Acetitomaculum*	−0.66 **	−0.40 *	−0.1	−0.2	0.52 *	−0.59 **	−0.22	−0.64 **
* norank_f_Eubacterium_coprostanoligenes_group*	−0.23	−0.01	0.33	−0.09	0.11	0.03	0.02	−0.22
* Christensenellaceae_R−7_group*	0.16	0.07	−0.14	−0.29	−0.03	0.26	−0.34	0.03
* Bifidobacterium*	−0.06	−0.21	−0.04	−0.19	0.42 *	−0.50 *	−0.15	0.03

* *p* < 0.05, ** *p* < 0.01. ^1^ NDF, neutral detergent fibre; ADF, acid detergent fibre; FAest, ester-linked ferulic acid; *p*CAest, ester-linked *p*-coumaric acid; *p*CAeth, ether-linked *p*-coumaric acid; FAeth, ether-linked ferulic acid; *p*CA/FA ratio, the ratio of *p*-coumaric acid to ferulic acid.

## Data Availability

The data are contained within the article.

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
