# Peer review of "In Situ Rumen Degradation Characteristics and Bacterial Colonization of Corn Silages Differing in Ferulic and p-Coumaric Acid Contents"

_microorganisms, 2022, doi:10.3390/microorganisms10112269_

Round 1

Reviewer 1 Report

The paper proposes collaborating to a better understanding of fiber degradation into the rumen and the role of lignin composes in these processes. However, the material and methods section is poorly written, and there are many important details that need to be included, for example, the treatment description and microbial statistical analysis. Other statistical approaches should be used as multivariate methods, to improve the answer that the authors want to provide.

L35 palatability? In ruminants certainly, this concept has evolved.

L42 what meaning PAs?

L45-44 “These linkages worked as a barrier to limit the utilization of plant cell walls.” Where?

L112 Larue's method?

There are several previous reports similar that are not considered as a background in the introduction. Some were used to support some results during the discussion.

There is no description of treatments, such as levels or concentrations of phenolic acids. Either information about corn silages, time of fermentation, specie or CV of maize, starch content, particle size, etc.

Why use 1 mm size for ruminal incubation, why not 2 mm?

Table 1 is not a results table, it should be shown in material and methods, and help to the description of the treatment.

The bacterial analysis is too poorly described, how many samples per silage and time were used for this analysis?

The DNA extraction even announced in a subtitle was not described. Bioinformatic details about quality trimming, database use to OTUs assigned, etc need to be described.

Statistical analyses of microbial data either are lost. I am really surprised because there are correlation results but nothing about it is detailed in the material and methods section. Taxonomic abundances are just described but there is no information about the significance or tendencies observed by treatments or incubation time.

Why did not evaluated fungi?

I am concerned about procedures during rinsed and squeezing bags, please give more details about it.

The cow effect was considered in mathematical models?

Author Response

Dear reviewer:

We appreciate you for your precious time in reviewing our paper and providing valuable comments. It was your valuable and insightful comments that led to possible improvements in the current version. We have carefully considered the comments and tried our best to address every one of them. All the revisal in the paper are highlighted in yellow . The point-to-point response are listed in the attachment.

We tried our best to improve the manuscript and made some changes. We really appreciate for your warm work earnestly, and hope the corrections will meet with approval. Angain thank you so much for your comments and suggestions.

 Best wishes!

Yanlu

Response to Reviewer 3 Comments

Dear Reviewer,

Thank you so much for the comments and constructive suggestions. Those comments are all valuable and very helpful for revising and improving our paper, as well as the important guiding significance to our researches. We have studied comments carefully and have made correction point to point which we hope meet with approval. Revised portions are marked in yellow in the paper. Finally, we really appreciate for your good comments.

General comments

In this study, the authors verify the rumen degradation and the variation of the main rumen bacterial and fungi populations, directly correlated to the different contents of ferulic and p-coumaric acid, contained in corn silage. The study is interesting, despite the use of rumen degradation method was used over the past, but today is still considered valid.

The manuscript in many parts presents serious inaccuracies and often, especially in the part of the results, is very heavy to read and understand. I suggest to rewrite the results part and make graphs and tables more clearly. In particular, I suggest summarizing the degradability study part a little (it is well known that the degradability is lower when the lignin content increases), and making the understanding more fluid. Personally, I would focus more on exposure and, in-depth, the analysis of ruminal populations, rather than on degradability. So I would summarize this last part.

Response 1: We appreciate you for your precious time in reviewing our paper and providing valuable comments. According to your advice, we give a more brief describe on the results and deleted some unessential parts about the rumen degradation. However, we are not able to deleted the lignin parts in the degradation forms. Because we hope to compare the effects between phenolic acids and lignin on rumen degradation. We give a more brief describe and rewrite the 3.2 part.

The introduction part is a little short. I suggest an implementation, in particular regarding the effects of ferulic and p-coumaric acids.

Response 2:Thanks for your advice. The corresponding part was added in Line 38-56

The scientific approach and the experimental protocol presents gaps in exposure, and raises some doubts about the experimental protocol used. I will provide some comments below. Overall, the data obtained from the study are hardly commented. In general, the manuscript also present a widespread errors in both grammar and form, including citation, space and font size. I recommend a complete grammar and English revision of the manuscript. The manuscript can be accepted after major revisions. Below, some examples of the errors, my comments and questions.

Response 3: Thanks for your advice. We have supplemented the describe on the Materials and Methods in line 131-174 and 163-170. We hope to the supplements could match your mind, and if any other revise you have, please tell us. We would like to try our best to make this paper more satisfied.

General errors:

Check all the text for double space and space between words and citations. Also, all abbreviations  (e.g. pCA, PAS, etc.) must be preceded by the extended wording.

Response 4: Sorry for this mistake. PAs in this passage means phenolic acids. And We have corrected in Line 48, 179 and 182

In the manuscript, the use of whole corn silage need to be better explained. In materials and methods it should be explained whether they are just different varieties or even different vegetative stages ... Were they at different vegetative stages? so consequently the lignin content increased? Explain in materials and methods.

Response 5: We are so sorry for this question. In order to get different phen0lic acids contented silages, we collected the silages from farms in different provinces. These corn silages were collected from different farms in China. The factory director did not provide us these information about the silages.

Q:Line 17: “whole corn silage” change with only “corn silage”. Correct in all parts of the text of the manuscript.

Response 6: Yes, accepted and revised

Q:Line 17-18: the method use, the name, is missing. It is essential indicate the method  (in situ rumen incubation) in the abstract. The method used and at least a mention of the animals used should be indicated in the abstract.

Response 7: Thanks for your advice. We added the method and the name in line 18 and line 19 with the highlight of yellow.

Line 21: cited: “, and the ratio pCA to FA (pCA/FA)”. Change in: “, and their ratio pCA/FA”.

Response 8: Yes, accepted and revised in line 23

Line 39: error in citation. The number 4 refers to: Jung and Vogel and not only Jung author. Please correct.

Response 9: Sorry for this mistake and we have corrected in line 42.

Line 58: “to determine”…change in “to evaluate”.

Response 10:Yes, accepted and revised in line 74.

Line 87:  “10 µL” correct in “ . Ten µL of sample was…..”.

Response 11: Yes, accepted and revised in line 109

Line 119: missing space.

Response 12:Yes, accepted and revised. Added a space between 5min and followed in Line 156.

Line 124: “imagennnanalysis”, correct in “image analysis

Response 13: Yes, accepted and revised in line159 and highlighted in yellow.

Line 134: citation error.

Response 14: Yes, accepted and revised in line 174

Line 138: citation error.

Response 15: :Yes, accepted and revised in line 177

Line 144: citation error.

Response 16: :Yes, accepted and revised in line 185

Line 150 citation error.

Response 17: Yes, accepted and revised in line 191

Line 164: “ after 72 h rumen fermentation…” correct in “ after 72 h of rumen fermentation”.

Response 18: Yes, accepted and revised in line 209,214 and 316.

In the text: “pearson” correct in “Pearson”.

Response 19: Yes, accepted and revised in line 245,275

Line 222: Rephrase, for example: “From this study,  a total of 18……. (Figure 1a)”..

Response 20: Yes, accepted and revised in line 257

Line 268, discussion part: “. And the value…….” Rephrase.

Response 21: Yes, accepted and revised in line 304

Line 268: with pCA/FA is necessary to write also “ratio”. The correct form is: “pCA/FA ratio”. Please correct in all part of the text (i.e, line 275).

Response 22: Yes, accepted and revised. All the correction has been highlighted in yellow.

Line at citation n. 24: “araboxylan”…is arabinoxylan?

Response 23: Yes, accepted and revised in line 318

Line 290: “So Giada pointed”…change in “So, Giada,  pointed….” Or “So, Giada [26], ….”. idem for citation n. 27.

Response 24: Yes, accepted and revised in line 325

Line 312: “Thus, in this paper”….Maybe, in this study? Verify the sense of this sentence.

Response 25: Yes, accepted and revised in line 347,461,and 422

In the text, the names of the bacteria or fungi should be written in italics.

Response 26: Yes, accepted and revised in line 275-277.

Line 336 to 338 (citation n. 41). Rephrase.

Response 27: Yes, accepted and revised in line 387

Line 339: “However, Javad also….” Delete also….”Javad suggested…”. This is the citation n. 12. However, the name of the autor is Javad but the surname is Gharechahi…Verify this citation.

Response 28: Sorry for the mistake. We have corrected in line 388.

 Meterials and Methods

Parag. 2.2. It should be explained why additional amylase is also used in the analytical method. Please add.

Response 28: In order to wash the starch in whole corn silages, we added amylase during the detection of NDF.

Parag. 2.4. The lactating cows used in this study, consumed 25 kg DM per day. It is mandatory indicate the lactation phase or, eventually, the production yield. They consumed “silage maize”….usually “maize silage”…. in this case, you intended the whole plant silage? What do you mean by mixture for animal?

Response 30: Thanks for your comments; we have revised this section again. The cow is in late- lactation phase. The term “Maize silage” has been revised as “corn silage”, the corn and mixture have been revised as commercial concentrate feed

Regarding the exposure of the samples, number of samples, and number of sample per animal used, everything must be rewritten. The mathematical count, do not add up.Based on the exposure, I don't understand if a bag was used for each time (6 timepoints excluding T0), and all were extracted .... How is it possible that 1 per animal remains for genetic analysis of rumen bacterial populations? Maybe I'm crazy!

Response 31: Sorry, there are something unclear written for the collection procedure. Now we have checked and rewritten the sampling procedure as showed in the section of ‘In situ rumen incubation’, ‘Bacterial community analysis’ as well as the section ‘Calculations and Statistical analysis’. Finally, the sampling procedure resulted 8 corn silages ´ 3 cows = 24 samples for genetic analysis of rumen bacterial populations.

Furthermore, it should be specified that the genetic investigations were performed on a sample at only 24 hours, it is not written in this part, but only later (Parag. 3.4). A coefficient of 0.0253 / h is chosen as the out flow rate. If possible, it would be necessary to justify the choice. Could the choice of this value have changed the outcome of your results?

Response 32: As explained in the above. We have added ‘(8 corn silages ´ 3 cows = 24 samples)’ in Parag. 3.4. Regarding the choice of coefficient of 0.0253 / h, it was used to calculate effective degradation (ED) consideration the outflow rate of rumen content to lower digestion tract, such a fixed constant is usually applied in rumen nutrition and feed evaluation, and it does not affect the overall outcome of the presents study since the difference of ED is mainly affected by the estimates of constant a, b, c  

 Results

In addition to the suggestions previously reported:

Can you explain why as lignin content increases, you do not have an increase in ferulic acids?

Response 32: During the formation of primary plant cell wall, ferulates act as nucleation sites for lignin formation[1,2]. Then lignin and arabinoxylans were connected by ferulate molecules through ether bonds and other linkages, and form dimeric structures that cross-link arabinoxylan chains to each other and to lignin [3]. Thus, the decrease of ferulic acid due to the conversion of ferulic acid to lignin.

The graphs are very little… better enlarge them.

Response 33: Thanks for your advice. All the figures in this study have been enlarged.

The discussion of the results is very similar to a review. In fact, other studies are mentioned without valid support and comment on their results.

Response 34: Thanks for your advice. We rewrote this part in line 319-324.

Furthermore, in many parts of the discussion, it is evident that the results obtained from your study are confused with those of other studies, with the simultaneous use of "in the present paper" or "in the present study". Please re-write these parts.

Response 35: Thanks for your advice. We have checked and all the revision are highlighted in yellow.

Reference:

  1. Ralph, J.; Hatfield, R.; Quideau, S.; Helm, R. Lignin crosslinking in the plant cell wall; unambiguous methods for identification, and structural/regiochemical characterization of cross-linked structures. In Proceedings of the Abstacts of papers of the American chemical society, 1993; pp. 137-CELL.
  2. Grabber, J.H.; Ralph, J.; Hatfield, R.D. Modeling lignification in grasses with monolignol dehydropolymerisate-cell wall complexes. Washington, DC: American Chemical Society: 1998; pp. 163-171.
  3. Ralph, J.; Quideau, S.; Grabber, J.H.; Hatfield, R.D. Identification and synthesis of new ferulic acid dehydrodimers present in grass cell walls. Journal of the Chemical Society, Perkin Transactions 1 1994, 3485-3498.

Reviewer 2 Report

The study presented is of good level,  the methods used well described, the  experimental design is appropriate and the results open interesting prospects of deepening.

Given the results, it is a pity that English is sometimes a little approximate, in my opinion more out of haste than lack of knowledge.

In my opinion, the study would also have benefited from having a greater variety in the quality of corn silage to accentuate the differences and draw even clearer conclusions.

Line 36 : "the present" please revise the english

Line 184-186 : Please revise like that "The ED of DM (r=-0.86, P<0.01),  pCAeth; r=-0.91, P<0.01), .....by correctly placing parentheses and punctuations.

Line 278 : losses instead of loss

Line 289 : please review english  "Different with FA,"

Line 301 : please review english "a measuirment"

Line 302 : please review english "and adherent"

Line 304 : please review english "was always showed"

Line 324 : please review english "could be explained that"

Line 374 : "more abundant" instead of "more enriched"

Line 375 : "positively correlated" instead of "positive related"

Line 391 : "was not impacted" instead of "did not effected"

Line 401-402 : please review the english "the content of lignin in forages was positive with the adherent of Prevotella"

Line 411 :  please review english "Olsenella mainly affected"

Line 416 : please review english "acetate and rich"

Line 418 : please prevent repetion  "increased with the increasing"

Line 426 : "adherence" instead of "adherent"

Line 435 : please review english "be more the influencing of attached bacteria"

Author Response

Dear reviewer,

First of all, great thanks for your time to give us review comments. Following your nice suggestion, we have check the whole manuscript point by point, and the revised places have been highlighted . Hopefully the revised version could be acceptable. All the corrections have been highlighted in yellow in the attachment.

Please see the attachment! Once again, thank you so much for your comments and suggestions.

Best wishes!

Yanlu

Line 36 : "the present" please revise the english

Response 1: Yes, accepted and revised in line 37

Line 184-186 : Please revise like that "The ED of DM (r=-0.86, P<0.01),  pCAeth; r=-0.91, P<0.01), .....by correctly placing parentheses and punctuations.

Response 2: Yes, accepted and revised in line 202-204

Line 278 : losses instead of loss

Response 3: Yes, accepted and revised in line 313

Line 289 : please review english  "Different with FA,"

Response 4: Yes, accepted and revised in line 324

Line 301 : please review english "a measuirment"

Response 5: Yes, accepted and revised in line 336

Line 302 : please review english "and adherent"

Response 6: Yes, accepted and revised in line 337

Line 304 : please review english "was always showed"

Response 7: Yes, accepted and revised in line 339

Line 324 : please review english "could be explained that"

Response 8: Yes, accepted and revised in line 359

Line 374 : "more abundant" instead of "more enriched"

Response 9: Yes, accepted and revised in line 409

Line 375 : "positively correlated" instead of "positive related"

Response 10: Yes, accepted and revised in line 410

Line 391 : "was not impacted" instead of "did not effected".

Response 11: Yes, accepted and revised in line 428

Line 401-402 : please review the english "the content of lignin in forages was positive with the adherent of Prevotella"

Response 12: Yes, accepted and revised in line 437-439

Line 411 :  please review english "Olsenella mainly affected"

Response 13: Yes, accepted and revised in line 448

Line 416 : please review english "acetate and rich"

Response 14: Yes, accepted and revised in line 454

Line 418 : please prevent repetion  "increased with the increasing"

Response 15: Yes, accepted and revised in line 464

Line 426 : "adherence" instead of "adherent"

Response 16: Yes, accepted and revised in line 465

Line 435 : please review english "be more the influencing of attached bacteria"

Response 17: Yes, accepted and revised in line 473

Reviewer 3 Report

General comments

In this study, the authors verify the rumen degradation and the variation of the main rumen bacterial and fungi populations, directly correlated to the different contents of ferulic and p-coumaric acid, contained in corn silage. The study is interesting, despite the use of rumen degradation method was used over the past, but today is still considered valid.

The manuscript in many parts presents serious inaccuracies and often, especially in the part of the results, is very heavy to read and understand. I suggest to rewrite the results part and make graphs and tables more clearly. In particular, I suggest summarizing the degradability study part a little (it is well known that the degradability is lower when the lignin content increases), and making the understanding more fluid. Personally, I would focus more on exposure and, in-depth, the analysis of ruminal populations, rather than on degradability. So I would summarize this last part.

The introduction part is a little short. I suggest an implementation, in particular regarding the effects of ferulic and p-coumaric acids.

The scientific approach and the experimental protocol presents gaps in exposure, and raises some doubts about the experimental protocol used. I will provide some comments below. Overall, the data obtained from the study are hardly commented. In general, the manuscript also present a widespread errors in both grammar and form, including citation, space and font size. I recommend a complete grammar and English revision of the manuscript. The manuscript can be accepted after major revisions. Below, some examples of the errors, my comments and questions.

 General errors:

Check all the text for double space and space between words and citations. Also, all abbreviations  (e.g. pCA, PAS, etc.) must be preceded by the extended wording.

In the manuscript, the use of whole corn silage need to be better explained. In materials and methods it should be explained whether they are just different varieties or even different vegetative stages ... Were they at different vegetative stages? so consequently the lignin content increased? Explain in materials and methods.

Line 17: “whole corn silage” change with only “corn silage”. Correct in all parts of the text of the manuscript.

Line 17-18: the method use, the name, is missing. It is essential indicate the method  (in situ rumen incubation) in the abstract. The method used and at least a mention of the animals used should be indicated in the abstract.

Line 21: cited: “, and the ratio pCA to FA (pCA/FA)”. Change in: “, and their ratio pCA/FA”.

Line 39: error in citation. The number 4 refers to: Jung and Vogel and not only Jung author. Please correct.

Line 58: “to determine”…change in “to evaluate”.

Line 87:  “10 µL” correct in “ . Ten µL of sample was…..”.

Line 119: missing space.

Line 124: “imagennnanalysis”, correct in “image analysis”.

Line 134: citation error.

Line 138: citation error.

Line 144: citation error.

Line 150 citation error.

Line 164: “ after 72 h rumen fermentation…” correct in “ after 72 h of rumen fermentation”.

In the text: “pearson” correct in “Pearson”.

Line 222: Rephrase, for example: “From this study,  a total of 18……. (Figure 1a)”.

Line 268, discussion part: “. And the value…….” Rephrase.

Line 268: with pCA/FA is necessary to write also “ratio”. The correct form is: “pCA/FA ratio”. Please correct in all part of the text (i.e, line 275).

Line at citation n. 24: “araboxylan”…is arabinoxylan?

Line 290: “So Giada pointed”…change in “So, Giada,  pointed….” Or “So, Giada [26], ….”. idem for citation n. 27.

Line 312: “Thus, in this paper”….Maybe, in this study? Verify the sense of this sentence.

In the text, the names of the bacteria or fungi should be written in italics.

Line 336 to 338 (citation n. 41). Rephrase.

Line 339: “However, Javad also….” Delete also….”Javad suggested…”. This is the citation n. 12. However, the name of the autor is Javad but the surname is Gharechahi…Verify this citation.

 Meterials and Methods

Parag. 2.2. It should be explained why additional amylase is also used in the analytical method. Please add.

Parag. 2.4. The lactating cows used in this study, consumed 25 kg DM per day. It is mandatory indicate the lactation phase or, eventually, the production yield. They consumed “silage maize”….usually “maize silage”…. in this case, you intended the whole plant silage? What do you mean by mixture for animal?

Regarding the exposure of the samples, number of samples, and number of sample per animal used, everything must be rewritten. The mathematical count, do not add up.

Based on the exposure, I don't understand if a bag was used for each time (6 timepoints excluding T0), and all were extracted .... How is it possible that 1 per animal remains for genetic analysis of rumen bacterial populations? Maybe I'm crazy!

Furthermore, it should be specified that the genetic investigations were performed on a sample at only 24 hours, it is not written in this part, but only later (Parag. 3.4). A coefficient of 0.0253 / h is chosen as the out flow rate. If possible, it would be necessary to justify the choice. Could the choice of this value have changed the outcome of your results?

 Results

In addition to the suggestions previously reported:

Can you explain why as lignin content increases, you do not have an increase in ferulic acids?

The graphs are very little… better enlarge them.

The discussion of the results is very similar to a review. In fact, other studies are mentioned without valid support and comment on their results. Furthermore, in many parts of the discussion, it is evident that the results obtained from your study are confused with those of other studies, with the simultaneous use of "in the present paper" or "in the present study". Please re-write these parts.

Author Response

Dear Reviewer,

Thank you so much for the comments and constructive suggestions. Those comments are all valuable and very helpful for revising and improving our paper, as well as the important guiding significance to our researches. We have studied comments carefully and have made correction point to point which we hope meet with approval. Revised portions are marked in yellow in the attachment. Please see the attachment! Finally, we really appreciate for your good comments. And thank you very much for your comments and suggestions.

Best wishes!

Yanlu

General comments

In this study, the authors verify the rumen degradation and the variation of the main rumen bacterial and fungi populations, directly correlated to the different contents of ferulic and p-coumaric acid, contained in corn silage. The study is interesting, despite the use of rumen degradation method was used over the past, but today is still considered valid.

The manuscript in many parts presents serious inaccuracies and often, especially in the part of the results, is very heavy to read and understand. I suggest to rewrite the results part and make graphs and tables more clearly. In particular, I suggest summarizing the degradability study part a little (it is well known that the degradability is lower when the lignin content increases), and making the understanding more fluid. Personally, I would focus more on exposure and, in-depth, the analysis of ruminal populations, rather than on degradability. So I would summarize this last part.

Response 1: We appreciate you for your precious time in reviewing our paper and providing valuable comments. According to your advice, we give a more brief describe on the results and deleted some unessential parts about the rumen degradation. However, we are not able to deleted the lignin parts in the degradation forms. Because we hope to compare the effects between phenolic acids and lignin on rumen degradation. We give a more brief describe and rewrite the 3.2 part.

The introduction part is a little short. I suggest an implementation, in particular regarding the effects of ferulic and p-coumaric acids.

Response 2:Thanks for your advice. The corresponding part was added in Line 38-56

The scientific approach and the experimental protocol presents gaps in exposure, and raises some doubts about the experimental protocol used. I will provide some comments below. Overall, the data obtained from the study are hardly commented. In general, the manuscript also present a widespread errors in both grammar and form, including citation, space and font size. I recommend a complete grammar and English revision of the manuscript. The manuscript can be accepted after major revisions. Below, some examples of the errors, my comments and questions.

Response 3: Thanks for your advice. We have supplemented the describe on the Materials and Methods in line 131-174 and 163-170. We hope to the supplements could match your mind, and if any other revise you have, please tell us. We would like to try our best to make this paper more

General errors:

Check all the text for double space and space between words and citations. Also, all abbreviations  (e.g. pCA, PAS, etc.) must be preceded by the extended wording.

Response 4: Sorry for this mistake. PAs in this passage means phenolic acids. And We have corrected in Line 48, 179 and 182

In the manuscript, the use of whole corn silage need to be better explained. In materials and methods it should be explained whether they are just different varieties or even different vegetative stages ... Were they at different vegetative stages? so consequently the lignin content increased? Explain in materials and methods.

Response 5: We are so sorry for this question. In order to get different phen0lic acids contented silages, we collected the silages from farms in different provinces. These corn silages were collected from different farms in China. The factory director did not provide us these information about the silages.

Q:Line 17: “whole corn silage” change with only “corn silage”. Correct in all parts of the text of the manuscript.

Response 6: Yes,accepted and revised

Q:Line 17-18: the method use, the name, is missing. It is essential indicate the method  (in situ rumen incubation) in the abstract. The method used and at least a mention of the animals used should be indicated in the abstract.

Response 7: Thanks for your advice. We added the method and the name in line 18 and line 19 with the highlight of yellow.

Line 21: cited: “, and the ratio pCA to FA (pCA/FA)”. Change in: “, and their ratio pCA/FA”.

Response 8: Yes, accepted and revised in line 23

Line 39: error in citation. The number 4 refers to: Jung and Vogel and not only Jung author. Please correct.

Response 9: Sorry for this mistake and we have corrected in line 42.

Line 58: “to determine”…change in “to evaluate”.

Response 10:Yes, accepted and revised in line 74.

Line 87:  “10 µL” correct in “ . Ten µL of sample was…..”.

Response 11: Yes, accepted and revised in line 109

Line 119: missing space.

Response 12:Yes, accepted and revised. Added a space between 5min and followed in Line 156.

Line 124: “imagennnanalysis”, correct in “image analysis

Response 13: Yes, accepted and revised in line159 and highlighted in yellow.

Line 134: citation error.

Response 14: Yes, accepted and revised in line 174

Line 138: citation error.

Response 15: :Yes, accepted and revised in line 177

Line 144: citation error.

Response 16: :Yes, accepted and revised in line 185

Line 150 citation error.

Response 17: Yes, accepted and revised in line 191

Line 164: “ after 72 h rumen fermentation…” correct in “ after 72 h of rumen fermentation”.

Response 18: Yes, accepted and revised in line 209,214 and 316.

In the text: “pearson” correct in “Pearson”.

Response 19: Yes, accepted and revised in line 245,275

Line 222: Rephrase, for example: “From this study,  a total of 18……. (Figure 1a)”..

Response 20: Yes, accepted and revised in line 257

Line 268, discussion part: “. And the value…….” Rephrase.

Response 21: Yes, accepted and revised in line 304

Line 268: with pCA/FA is necessary to write also “ratio”. The correct form is: “pCA/FA ratio”. Please correct in all part of the text (i.e, line 275).

Response 22: Yes, accepted and revised. All the correction has been highlighted in yellow.

Line at citation n. 24: “araboxylan”…is arabinoxylan?

Response 23: Yes, accepted and revised in line 318

Line 290: “So Giada pointed”…change in “So, Giada,  pointed….” Or “So, Giada [26], ….”. idem for citation n. 27.

Response 24: Yes, accepted and revised in line 325

Line 312: “Thus, in this paper”….Maybe, in this study? Verify the sense of this sentence.

Response 25: Yes, accepted and revised in line 347,461,and 422

In the text, the names of the bacteria or fungi should be written in italics.

Response 26: Yes, accepted and revised in line 275-277.

Line 336 to 338 (citation n. 41). Rephrase.

Response 27: Yes, accepted and revised in line 387

Line 339: “However, Javad also….” Delete also….”Javad suggested…”. This is the citation n. 12. However, the name of the autor is Javad but the surname is Gharechahi…Verify this citation.

Response 28: Sorry for the mistake. We have corrected in line 388.

 Meterials and Methods

Parag. 2.2. It should be explained why additional amylase is also used in the analytical method. Please add.

Response 28: In order to wash the starch in whole corn silages, we added amylase during the detection of NDF.

Parag. 2.4. The lactating cows used in this study, consumed 25 kg DM per day. It is mandatory indicate the lactation phase or, eventually, the production yield. They consumed “silage maize”….usually “maize silage”…. in this case, you intended the whole plant silage? What do you mean by mixture for animal?

Response 30: Thanks for your comments, we have revised this section again. The cow is in late- lactation phase. The term “Maize silage” has been revised as “corn silage”, the corn and mixture have been revised as commercial concentrate feed

Furthermore, it should be specified that the genetic investigations were performed on a sample at only 24 hours, it is not written in this part, but only later (Parag. 3.4). A coefficient of 0.0253 / h is chosen as the out flow rate. If possible, it would be necessary to justify the choice. Could the choice of this value have changed the outcome of your results?

Response 31: We explained the sample on detected after 24 h of rumen fermentation in line 135. Sorry for the mistake, and I have added the corresponding reference about the 0.0253/h in line 194。

 Results

In addition to the suggestions previously reported:

Can you explain why as lignin content increases, you do not have an increase in ferulic acids?

Response 32: During the formation of primary plant cell wall, ferulates can act as nucleation sites for lignin formation[1,2]. Then lignin and arabinoxylans were connected by ferulate molecules through ether bonds and other linkages, and form dimeric structures that cross-link arabinoxylan chains to each other and to lignin [3]. Thus, the decrease of ferulic acid due to the conversion of ferulic acid to lignin.

The graphs are very little… better enlarge them.

Response 33: Thanks for your advice. All the figures in this study have been enlarged.

The discussion of the results is very similar to a review. In fact, other studies are mentioned without valid support and comment on their results.

Response 34: Thanks for your advice. We rewrote this part in line 319-324.

Furthermore, in many parts of the discussion, it is evident that the results obtained from your study are confused with those of other studies, with the simultaneous use of "in the present paper" or "in the present study". Please re-write these parts.

Response 35: Thanks for your advice. We have checked and all the revision are highlighted in yellow.

Reference:

  1. Ralph, J.; Hatfield, R.; Quideau, S.; Helm, R. Lignin crosslinking in the plant cell wall; unambiguous methods for identification, and structural/regiochemical characterization of cross-linked structures. In Proceedings of the Abstacts of papers of the American chemical society, 1993; pp. 137-CELL.
  2. Grabber, J.H.; Ralph, J.; Hatfield, R.D. Modeling lignification in grasses with monolignol dehydropolymerisate-cell wall complexes. Washington, DC: American Chemical Society: 1998; pp. 163-171.
  3. Ralph, J.; Quideau, S.; Grabber, J.H.; Hatfield, R.D. Identification and synthesis of new ferulic acid dehydrodimers present in grass cell walls. Journal of the Chemical Society, Perkin Transactions 1 1994, 3485-3498.

Round 2

Reviewer 3 Report

The authors have made the necessary revisions, making the manuscript clearer and more fluid. However, I report some minor (probably distraction) revisions.

Line 26: Firmicutes

Line 64: error of form in the citation and in "and found".

Line 91-92: The reason for the use of amylase was not reported.

Line 108: errors in international units and spaces.

Line 132-133: pay attention to the spaces and titles of the paragraphs in italics.

Line 142: methanol, and ......

Table 4: Error in phylum ....

Good luck for the publication process!

Author Response

Dear Reviewer,

First of all, great thanks for your time to give us review comments. Following your nice suggestion, we have check the whole manuscript point by point, and the revised places have been highlighted . Hopefully the revised version could be acceptable. All the corrections have been highlighted in green.

Best wishes!

Yanlu Wang

The authors have made the necessary revisions, making the manuscript clearer and more fluid. However, I report some minor (probably distraction) revisions.

Line 26: Firmicutes

Response 1: Thanks for your comments and we have revised in line 26 and highlighted in green.

Line 64: error of form in the citation and in "and found".

Response 2:Thanks for your advice and we have revised in line 64 and highlighted in green.

Line 91-92: The reason for the use of amylase was not reported.

Response 3:Thanks for your advice and we have added the reason for the use of amylase in line 93 and highlighted in green.

Line 108: errors in international units and spaces.

Response 4:Thanks for your advice and we have revised in line 108

Line 132-133: pay attention to the spaces and titles of the paragraphs in italics.

Response 5:Thanks for your advice and we have revised in line 132-133

Line 142: methanol, and ......

Response 6:Thanks for your advice and we have revised in line 138

Table 4: Error in phylum ....

Response 7:Sorry for the mistake and we have revised in Table 4.

Good luck for the publication process!